# GrassmannTN: a Python package for Grassmann tensor network computations

Atis YOSPRAKOB*

*Department of Physics, Niigata University, Niigata 950-2181, Japan*

**Abstract**

We present GrassmannTN, a Python package for the computation of the Grassmann tensor network. The package is built to assist in the numerical computation without the need to input the fermionic sign factor manually. It prioritizes coding readability by designing every tensor manipulating function around the tensor subscripts. The computation of the Grassmann tensor renormalization group and Grassmann isometries using GrassmannTN are given as the use case examples.

---

*E-mail address : ayosp(at)phys.sc.niigata-u.ac.jp

# 1 Introduction

In theoretical physics, several problems demand the computation of quantities involving multi-variable integrals or summations. Examples include the path integral, thermal partition function, and calculation of low-lying states in quantum many-body systems. These quantities are often unsolvable using analytical methods, necessitating the use of computers for accurate results. However, a major challenge arises when dealing with a large number of degrees of freedom, as the complexity of the summation becomes difficult to handle. In such cases, tensor networks offer a solution. For instance, let's consider an $n$-particle wave

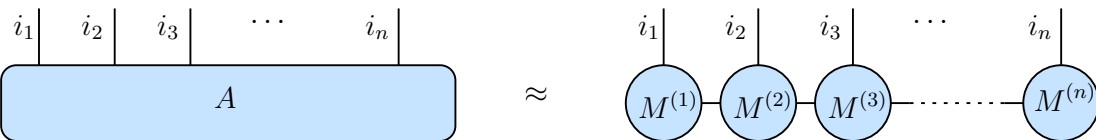

Figure 1: The matrix product state representation of the $n$-particle wave function (1.2).

function

$$|\Phi\rangle = \sum_{i_1,i_2,\cdots,i_n} A_{i_1i_2\cdots i_n}|i_1\rangle \otimes |i_2\rangle \otimes \cdots \otimes |i_n\rangle. \tag{1.1}$$

In this equation, $|i_a\rangle$ denotes a basis for a single particle state. Assuming that each $|i_a\rangle$ belongs to a $D$-dimensional Hilbert space, the coefficient tensor $A_{i_1i_2\cdots i_n}$ consists of $D^n$ individual components. In a realistic case where the number of particles $n$ is large, any computation involving this wave function will require the resource and time to grow exponentially with $n$. Such a heavy computation can be drastically reduced if we approximate the coefficient tensor in (1.1) by a product of order-3 tensors, known as the matrix product state (MPS) representation

$$A_{i_1i_2\cdots i_n} \approx \sum_{j_1=1}^{\chi}\sum_{j_2=1}^{\chi}\cdots\sum_{j_{n-1}=1}^{\chi} M_{i_1j_1}^{(1)} M_{j_1i_2j_2}^{(2)} M_{j_2i_3j_3}^{(3)} \cdots M_{j_{n-1}i_n}^{(n)}. \tag{1.2}$$

In other words, we rewrite the tensor $A_{i_1i_2\cdots i_n}$ in terms of a network of sub-tensors $M_{j_{a-1}i_aj_a}^{(a)}$. The diagrammatic representation of the MPS is shown in figure 1. Here, the auxiliary indices $j_a$ are all restricted to be of dimension $\chi$. Each of the sub-tensors consists of at most $D\chi^2$ components, which means that we only need at most $nD\chi^2$ degrees of freedom to represent the wave function. In many systems, even with a small $\chi$, this approximation often yields satisfactory results [1, 2, 3, 4]. Thus, the tensor network allows us to extract the essential physics of complex systems with a small computational resource [4, 5, 6, 7, 8, 9, 10].

Another application of the tensor network technique is in the computation of a partition function or a path integral, which typically takes the form

$$Z = \prod_{\vec{n}\in\Lambda} \sum_{x_{\vec{n}}} e^{-S_{\vec{n}}[x]}. \tag{1.3}$$

Here, we assume that the degrees of freedom $x_{\vec{n}}$ are located at the site $\vec{n}$ on the $d$-dimensional hyper-cubic lattice $\Lambda$. Using an appropriate transformation, the partition function can be rewritten in terms of 'link variables' instead[1]

$$Z = \prod_{\vec{n}\in\Lambda} \sum_{u_{\vec{n},1}} \cdots \sum_{u_{\vec{n},d}} e^{-S'_{\vec{n}}[u]}. \tag{1.4}$$

---

[1]See Ref. [11, 12, 13, 14] for some examples.

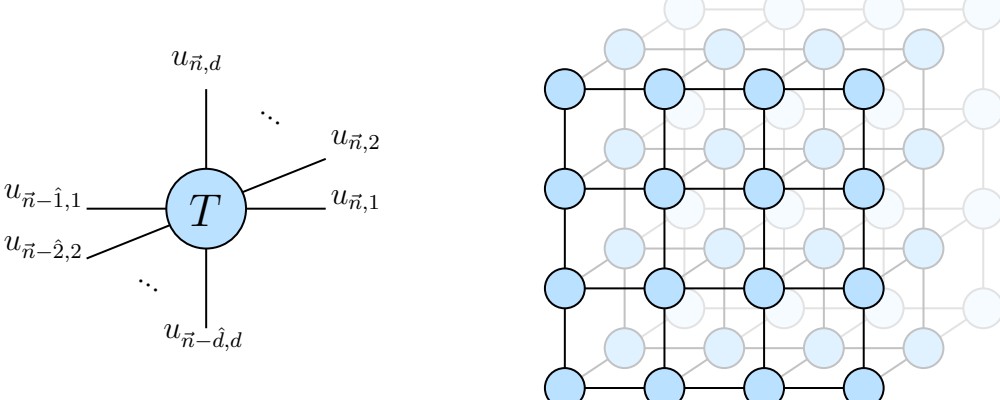

Figure 2: (Left) The Boltzmann weight based on link variables (1.5). (Right) The three-dimensional ($d = 3$) tensor network.

The link variable $u_{\vec{n},\mu}$ is a degree of freedom that is located on the link between the site $\vec{n}$ and $\vec{n} + \hat{\mu}$. If the system is highly localized, the action $S'_{\vec{n}}[u]$ will depend only on link variables surrounding the site $\vec{n}$, which means that we can write

$$e^{-S'_{\vec{n}}[u]} = T_{u_{\vec{n},1}, u_{\vec{n}-\hat{1},1}, u_{\vec{n},2}, u_{\vec{n}-\hat{2},2}, \cdots u_{\vec{n},d}, u_{\vec{n}-\hat{d},d}}, \tag{1.5}$$

which is often called the 'site tensor'. It can be depicted diagrammatically as in figure 2.

In this equation, the link variables act as the tensor indices of the Boltzmann weight. In this form, the partition function (1.4) is essentially a tensor network since the summation of all link variables acts as the contraction of tensor legs. The boon of representing the partition function as a tensor network is that it allows us to perform a coarse-graining procedure, which approximates the original tensor network by a new network with a smaller number of degrees of freedom. After a sufficient number of coarse-graining iterations, the partition function can be reduced to a trace of a single tensor. This class of algorithm is generically known as the tensor renormalization group (TRG) approach. The first version of the TRG algorithm applies to a two-dimensional bosonic spin system [11]. The improved versions had been subsequently proposed [15, 16, 17]. It can also be generalized to higher dimensional lattice [18, 19, 20, 21]. Most importantly, the partition function with fermionic or Grassmann degrees of freedom can be dealt with directly without the need to integrate the fermions out first [22, 23, 24, 14, 21]. Recently, the TRG has been applied to gauge theories and strongly correlated fermionic systems [25, 26, 27, 28, 24, 29, 30, 31, 32, 33, 34, 13], which shows that it is a promising approach aside from the Monte Carlo methods.

Before the development of the Grassmann tensor network, fermions must be bosonized in one way or another. For example, to describe a fermionic state via the ansatz state, the

fermionic operators are first transformed using the Jordan-Wigner transformation into spin operators [35] (see also Ref. [36, 37, 38, 39, 40] for its application to well-known tensor network states.) In the Monte Carlo treatment of the lattice gauge theory, the fermions are first integrated into the determinant:

$$Z = \int \mathcal{D}U \int \mathcal{D}\bar{\psi}\mathcal{D}\psi \, e^{-S[U]+\bar{\psi}\slashed{D}[U]\psi} = \int \mathcal{D}U \, \det \slashed{D}[U] \, e^{-S[U]}. \qquad (1.6)$$

The determinant is then treated as a part of the Boltzmann weight. However, the determinant $\det \slashed{D}[U]$ is known to be computationally demanding since the fermion matrix size grows like a power law of the system size. Such fermionic degrees of freedom can be treated directly with the introduction of Grassmann tensors [22, 23]. In the Grassmann tensor renormalization group (gTRG) methods, the partition function can be computed with the logarithmic complexity of the system size, which allows us to access the thermodynamic limit significantly more easily. Similarly to the bosonic TRG methods, we first transform the site fermions $\psi_{\vec{n}}$ into link fermions $\eta_{\vec{n},\mu}$, and then rewrite the Boltzmann weight as a Grassmann tensor

$$\mathcal{T}_{\eta_{\vec{n},1},\bar{\eta}_{\vec{n}-\hat{1},1},\eta_{\vec{n},2},\bar{\eta}_{\vec{n}-\hat{2},2},\cdots\eta_{\vec{n},d},\bar{\eta}_{\vec{n}-\hat{d},d}}. \qquad (1.7)$$

Coarse-graining algorithms similar to those for the non-Grassmann case can then be applied. The numerical computations on the Grassmann tensors are done through the coefficient tensors $T$;

$$\mathcal{T}_{\eta_{\vec{n},1},\bar{\eta}_{\vec{n}-\hat{1},1},\cdots\eta_{\vec{n},d},\bar{\eta}_{\vec{n}-\hat{d},d}} = \sum_{I_1,J_1,\cdots,I_d,J_d} T_{I_1 J_1\cdots I_d J_d}\eta_{\vec{n},1}^{I_1}\bar{\eta}_{\vec{n}-\hat{1},1}^{J_1}\cdots\eta_{\vec{n},d}^{I_d}\bar{\eta}_{\vec{n}-\hat{d},d}^{J_d}. \qquad (1.8)$$

Here, the indices $I_a$ and $J_a$ can be considered as the 'occupation number' of the link fermions. Although the Grassmann tensors $\mathcal{T}$ are not complex-valued, the coefficient tensors $T$ are and thus can be worked out on the computer.

One thing to keep in mind when working with Grassmann numbers is that fermions are anti-commuting. This means that the relative position of the fermions in (1.8) are very important, as they will affect the sign factors. One can already notice that even with a simple operation such as tensor contraction, many preparatory actions must be taken care of first. This is even more so with more complicated operations such as the gTRG algorithms. On the programming side, a Grassmann tensor contains more information than just the numerical values of the coefficient tensors. Managing the information in a clear and systematic way can be challenging when there are many fermions involved in the operation.

Here, we present a Python package `grassmanntn` that aims to address all of these issues. Firstly, the sign factors are implicitly computed in every computation. Secondly, every function is designed to work with tensor subscripts as the input, making the code easily

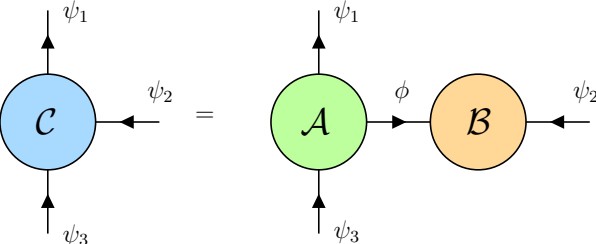

Figure 3: Diagrammatic representation of Grassmann tensor contraction (2.1) (also (A.26)).

translated from the symbolic expression. The usefulness of the package is demonstrated with the computation of the Levin-Nave TRG method and the computation of isometry tensors. The first application of the package is the study of the lattice gauge theory with multiple fermion flavors [13], which successfully reproduced known results as well as demonstrated the Silver Blaze phenomenon. The package is available online on the GitHub repository [41].

The rest of this paper is organized as follows. We first explain the design principles of `grassmanntn` in section 2. Section 3 discusses the main features of the package. Two coding examples are given in section 4. Section 5 is devoted to the summary and discussion. The mathematical formulation for the Grassmann tensor network is given in appendix A.

## 2 Design principles

The biggest obstacle in the numerical computation involving Grassmann tensors is the sign factor arising in various steps of the algebraic manipulation such as index swapping, index joining and splitting, and tensor contraction. Dealing with these sign factors requires additional blocks of code that the programmer has to write manually. This requires a lot of attention, especially for complex tasks like implementing tensor renormalization group algorithms where mistakes can easily occur. To give an example, a Grassmann contraction[2]

$$\mathcal{C}_{\psi_1\bar\psi_2\bar\psi_3} = \int_{\bar\phi\phi} \mathcal{A}_{\psi_1\phi\bar\psi_3}\mathcal{B}_{\bar\psi_2\bar\phi} \tag{2.1}$$

can be computed via the following coefficient contraction:

$$C_{IJK} = \sum_L A_{ILK}B_{JL}s_{JKL} \tag{2.2}$$

with a sign factor tensor (see (A.26))

$$s_{JKL} = \sigma_L \times (-)^{p(L)(p(J)+p(K))+p(J)p(K)} \tag{2.3}$$

---

[2]See appendix A for definitions and notations.

where $\sigma_I$ is a sign factor given in (A.13). This sign factor is composed of those from fermion anti-commutation and contraction. To code this in Python with the `numpy` package [42], the parity function $p(I)$ (A.9) and $\sigma_I$ are first defined:

```python
>>> def gparity(I):
...     # This is p(I) where I is a composite index
...     # Convert I (canonically encoded) to binary
...     I_binary = [ int(c) for c in format(I,'b') ]
...     return sum(I_binary)
...
>>> def sgn(I):
...     # This is sigma_I
...     I_binary = [ int(c) for c in format(I,'b') ]
...     n_bits = len(I_binary)
...     s = 1
...     for a in range(1,n_bits):
...         for b in range(a):
...             s *= (-1)**(I_binary[a]*I_binary[b])
...     return s
```

Then the sign factor tensor (2.3) is constructed:

```python
>>> import numpy as np
>>> nI, nL, nK = A.shape # Obtaining A's index dimensions
>>> nJ, nL = B.shape      # Obtaining B's index dimensions
>>> sign_factor = np.zeros( [nJ,nK,nL], dtype=int ) # The sign factor
>>> for J in range(nJ):
...     for K in range(nK):
...         for L in range(nL):
...             sign_factor[J,K,L] = sgn(L)*(-1)**(
...                 gparity(L)*(gparity(J)+gparity(K))+gparity(J)*gparity(K))
```

And finally, the contraction:

```python
>>> C = np.einsum('ILK,JL,JKL->IJK',A,B,sign_factor)
```

The function $p(I)$ and $\sigma_I$ can be reused in other contractions, but the sign factor (2.3) must be recalculated and rewritten for every contraction. It is not difficult to see that this can be arduous and is prone to mistakes as the program becomes more complex.

The first goal of the `grassmanntn` package is to eliminate the need for the user to compute these sign factors manually. In order to do that, `grassmanntn` introduces the Grassmann tensor as a programming object that contains information about the indices as well as the coefficient tensor. All functions will make use of this information to compute the sign factors implicitly—reducing the user input to the minimum.

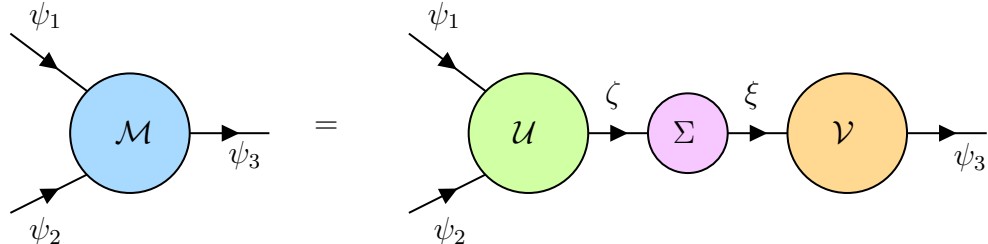

Figure 4: Diagrammatic representation of the singular value decomposition (2.4)

The second goal is to implement the functions with a declarative programming philosophy, where the user only has to tell the program what they want instead of how to obtain the result. For example, the Grassmann tensor contraction in the previous example can be computed with the `grassmanntn.einsum` function:

```
1 >>> import grassmanntn as gtn
2 >>> C = gtn.einsum('ILK,JL->IJK',A,B)
```

Similar to `numpy.einsum`, the only input the user has to enter is the subscripts of the operands, where the repeated characters are contracted. The properties of the resulting tensor, such as shape and index statistics, are determined automatically. The package also provides other operations such as complex conjugation, index joining and splitting, singular value decomposition (SVD), and eigenvalue decomposition (EigD), among others.

An upshot for this programming design is it is straightforward to write the code from the symbolic expression. For example, the tensor $\mathcal{M}_{\bar{\psi}_1\bar{\psi}_2\psi_3}$ can be decomposed with an SVD as

$$\mathcal{M}_{\bar{\psi}_1\bar{\psi}_2\psi_3} = \int_{\bar{\zeta}\zeta}\int_{\bar{\xi}\xi}\mathcal{U}_{\bar{\psi}_1\bar{\psi}_2\zeta}\Sigma_{\bar{\zeta}\xi}\mathcal{V}_{\bar{\xi}\psi_3} \tag{2.4}$$

where $\Sigma$ is a diagonal singular value matrix (see section A.7). This can be computed with the following code:

```
1 >>> # Initialize a random Grassmann tensor
2 >>> M = gtn.random( shape=(4,4,4), statistics=(-1,-1,1) )
3 >>> # Performing the singular value decomposition
4 >>> U,S,V = M.svd('IJ|K')          # SVD: U[I,J,A], S[A,B], and V[B,K]
```

Here, the SVD is performed between the first two Grassmann indices and the third, which is represented by the string `IJ|K`.

The tensors $\mathcal{U}$ and $\mathcal{V}$ are unitary; i.e.,

$$\int_{\bar{\psi}_1\psi_1}\int_{\bar{\psi}_2\psi_2}\mathcal{U}^{\dagger}_{\bar{\zeta}\psi_1\psi_2}\mathcal{U}_{\bar{\psi}_1\bar{\psi}_2\xi} = \int_{\bar{\psi}_3\psi_3}\mathcal{V}_{\bar{\zeta}\psi_3}\mathcal{V}^{\dagger}_{\psi_3\xi} = \mathcal{I}_{\bar{\zeta}\xi}, \tag{2.5}$$

where the identity Grassmann matrix is defined in (A.39). The following code demonstrates that $\mathcal{U}$ and $\mathcal{V}$ are unitary:

```
>>> Udagger = U.hconjugate('IJ|A') # Complex conjugate: Udagger[A,I,J]
>>> Vdagger = V.hconjugate('B|K')  # Complex conjugate: Vdagger[K,B]
>>> I1 = gtn.einsum('AIJ,IJB->AB',Udagger,U) # Udagger*U
>>> I2 = gtn.einsum('AK,KB->AB',V,Vdagger)   # V*Vdagger
>>> I1.force_format('matrix').display() # Show the coefficient elements

  array type: dense
        shape: (4, 4)
      density: 4 / 16 ~ 25.0 %
   statistics: (-1, 1)
       format: matrix
      encoder: canonical
       memory: 296 B
         norm: 2.0
      entries:
    (0, 0) 1.0
    (1, 1) 1.0
    (2, 2) 1.0
    (3, 3) 1.0

>>> I2.force_format('matrix').display() # Show the coefficient elements

  array type: dense
        shape: (4, 4)
      density: 4 / 16 ~ 25.0 %
   statistics: (-1, 1)
       format: matrix
      encoder: canonical
       memory: 296 B
         norm: 2.0
      entries:
    (0, 0) 1.0
    (1, 1) 1.0
    (2, 2) 1.0
    (3, 3) 1.0
```

The result shows that both $\mathcal{U}^\dagger\mathcal{U}$ and $\mathcal{V}\mathcal{V}^\dagger$ give a $4 \times 4$ identity matrix.

The package grassmanntn can be downloaded from the online repository [41]. The web documentation for grassmanntn is provided[3] where each class, function, and module are

---

[3]https://ayosprakob.github.io/grassmanntn/

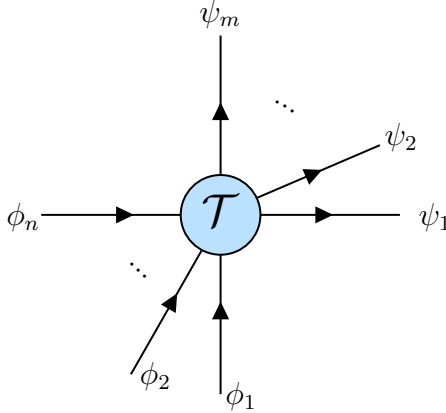

Figure 5: A Grassmann tensor of order $(m, n)$; $\mathcal{T}_{\psi_1 \cdots \psi_m \bar{\phi}_1 \cdots \bar{\phi}_n}$.

described in detail, with useful examples.

# 3 Features

In this section, we explain the main features of the package `grassmanntn` as of build `1.2.3`. Full details are given on the web documentation. For the mathematical formulation of the Grassmann tensor network, see appendix A.

## 3.1 Grassmann tensors as a programming object

Every Grassmann tensor

$$\mathcal{T}_{\psi_1 \cdots \psi_m \bar{\phi}_1 \cdots \bar{\phi}_n} = \sum_{I_1, \cdots, I_m, J_1, \cdots, J_n} T_{I_1 \cdots I_m J_1 \cdots J_n} \psi_1^{I_1} \cdots \psi_m^{I_m} \bar{\phi}_1^{J_1} \cdots \bar{\phi}_n^{J_n} \tag{3.1}$$

contains 4 kinds of information: the numerical coefficient tensor $T$, the statistics of the indices, the index encoding method, and the coefficient format; all of which are explained below.

*Statistics* refers to the type of index which can be: $+1$ for a non-conjugate fermionic index, -1 for a conjugated index, and 0 for a bosonic index. Diagrammatically, the non-conjugated fermionic index corresponds to the tensor leg with an arrow pointing away from the tensor, the conjugated index corresponds to the leg with an arrow pointing into the tensor, while bosonic legs do not have the arrow. An example of a tensor with $m$ non-conjugated legs and $n$ conjugated legs (3.1) is shown in figure 5.

*Index encoder* refers to how the composite index $I = (i_1, \cdots, i_n)$ is encoded as an

integer. There are two options, *canonical* and the *parity-preserving* [14] encoders:

$$I_{\text{canonical}}(i_1, \cdots, i_n) = \sum_{k=1}^{n} 2^{k-1} i_k, \tag{3.2}$$

$$I_{\text{parity-preserving}}(i_1, \cdots, i_n) = \begin{cases} \displaystyle\sum_{k=1}^{n} 2^{k-1} i_k & ; i_2 + \cdots + i_n \text{ even}, \\ 1 - i_1 + \displaystyle\sum_{k=2}^{n} 2^{k-1} i_k & ; i_2 + \cdots + i_n \text{ odd}. \end{cases} \tag{3.3}$$

The canonical encoder has the advantage that it is easy to join and split indices. For example, if $I$ and $J$ corresponds to the canonical indices of an $m$-bit fermion and an $n$-bit fermion, respectively, then $I$ and $J$ can be joined with

$$K = I + 2^m J \quad \text{(with canonical encoder)}, \tag{3.4}$$

which corresponds to $(i_1, \cdots, i_m, j_1, \cdots, j_n)$ in the bit representation. The parity-preserving encoder, as the name suggests, is designed in a way that the Grassmann parity of the index is readily manifested. Namely, if $I$ is a parity-preserving index corresponds to $(i_1, \cdots, i_n)$ in the bit representation, then we have

$$I \equiv \sum_{a=1}^{n} i_a \,(\text{mod } 2) \quad \text{(with parity-preserving encoder)}. \tag{3.5}$$

The two encoders can be switched by the switching function

$$\varepsilon(I_{\text{canonical}}) = \varepsilon^{-1}(I_{\text{canonical}}) = I_{\text{parity-preserving}}, \tag{3.6}$$

which is self-inverse. The encoder switching function can be accessed via the function `grassmanntn.param.encoder(I)`, where `I` is the encoded index to be switched.

The *coefficient format* refers to whether the coefficient tensor is in the *standard* or the *matrix format*, which are explained in detail in appendix A.6.

The package `grassmanntn` processes all of this information in a single programming object: `grassmanntn.dense` or `grassmanntn.sparse`, depending on whether the coefficient is stored in a sparse or dense format. Although the algorithms for the sparse and dense tensors are different, the two objects can be used together, where the package will choose the appropriate algorithm automatically.

`grassmanntn.dense` is built upon the dense multidimensional array `numpy.ndarray` from the `numpy` package [42] while `grassmanntn.sparse` is built upon a sparse array `sparse.COO` from the `sparse` package [43]. The coefficient tensor, the index statistics, the encoder, and the coefficient format can be accessed as the attributes of the object.

**Examples**

To make a random dense Grassmann tensor $\mathcal{T}_{\psi\bar{\phi}\bar{\zeta}mn}$ where $m$ and $n$ are bosonic indices with dimensions $d_\psi = d_\phi = 4$, $d_\zeta = 8$, and $d_m = d_n = 5$, the following command is used:

```
>>> import numpy as np
>>> import grassmanntn as gtn
>>> T_data = np.random.rand(4,4,8,5,5) # a random coeff with
>>>                                     # the specified shape.
>>> T_statistics = (1,-1,-1,0,0)        # the statistics of the indices.
>>> T_dense = gtn.dense(  data=T_data, statistics=T_statistics,
...                     encoder="canonical", format="standard")
```

Alternatively, the `grassmanntn.random()` function can also be used to generate a random Grassmann tensor:

```
>>> T_dense = gtn.random( shape=(4,4,8,5,5), statistics=(1,-1,-1,0,0),
...                     tensor_type=gtn.dense, dtype=float,
...                     encoder="canonical", format="standard",
...                     skip_trimming=True) # If False (default),
>>>                                         # the Grassmann-odd components
>>>                                         # are removed.
```

Sparse Grassmann tensor can also be initialized in the COO (coordinate list) format if a list of non-zero entries is specified. For example, if one wants to initialize the following tensor (canonical and standard):

$$\mathcal{T}_{\bar{\psi}\phi} = 3.1\bar{\psi}^3\phi^5 + (7.9 + 2.3i)\bar{\psi}^2\phi^7 + 5.8\bar{\psi}^0\phi^1 - 0.2i\bar{\psi}^2\phi^2 \tag{3.7}$$

where $\bar{\psi}$ and $\phi$ are 2- and 3-bit fermions ($d_\psi = 4$ and $d_\phi = 8$), respectively, then we write

```
>>> import sparse as sp
>>> cI = complex(0,1)
>>> T_shape = (4,8)
>>> T_statistics = (-1,1)
>>> psi_bar = [3, 2, 0, 2]                     #psi_bar's index
>>> phi     = [5, 7, 1, 2]                     #phi's index
>>> coords  = [psi_bar,phi]
>>> coeff   = [3.1, 7.9+2.3*cI, 5.8, -0.2*cI] #the coefficients
>>> T_data = sp.COO(coords,coeff,shape=T_shape)
>>> T_sparse = gtn.sparse( data=T_data, statistics=T_statistics,
...                     encoder="canonical", format="standard")
```

The two formats can be easily converted with

```
>>> T_dense_to_sparse = gtn.sparse(T_dense) # from dense to sparse
>>> T_sparse_to_dense = gtn.dense(T_sparse) # from sparse to dense
```

The coefficient of the Grassmann tensor as a multi-dimensional array can be extracted via the property `data`:

```
1 >>> data_dense  = T_dense.data
2 >>> data_sparse = T_sparse.data
3 >>> print( '  dense data type:', type(data_dense),
4 ...           '\n sparse data type:', type(data_sparse) )
5   dense data type: <class 'numpy.ndarray'>
6  sparse data type: <class 'sparse._coo.core.COO'>
```

## 3.2  Tensor contraction

Contractions between two indices can be done if 1) they have the same dimensions and 2) their statistics are the opposite. This includes the usual bosonic contraction and the fermionic contraction. In `grassmanntn` , contractions can be done via `einsum()`, which is designed to work in a similar way with `numpy.einsum()`.

The function `grassmanntn.einsum()` is built upon the highly optimized contraction function `opt_einsum.contract()` which works for both dense and sparse format of the coefficient tensor. As of `grassmanntn 1.2.3`, the bottleneck of the computation time comes from the sign factor computation, which we plan to improve in future versions.

**Examples**

First, prepare some tensors:

```
1 >>> import grassmanntn as gtn
2 >>> T = gtn.random(shape=(4,8,4),statistics=(1,-1,-1))
3 >>> S = gtn.random(shape=(4,4,6),statistics=(1,-1,0))
4 >>> G = gtn.random(shape=(6,6),statistics=(0,0))
5 >>> M = gtn.random(shape=(8,8),statistics=(1,-1))
```

The function `einsum()` can be used to perform a contraction within the same object

```
1 >>> v = gtn.einsum('iji->j',T)
2 >>> print('shape=',v.shape,', stats=',v.statistics)
3 shape= (8,) , stats= (-1,)
```

In this example, we perform the contraction between the first and the third index of `T`, leaving only the second index as a free index. This is described by the first argument, 'iji->j' where the repeated index `i` are summed (in the Einstein notation). The right-hand side of `->` indicates the indices of the result. Unless the contraction is complete, the right-hand side of `->` **must** be specified in order to get the intended result.

The indices can also be rearranged with `einsum`:

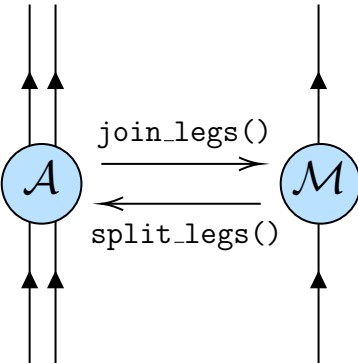

Figure 6: Diagrammatic representation of the reshaping process between the order-4 tensor A and the order-2 tensor M. Legs with an arrow pointing away from the tensor have the +1 statistics while legs with an arrow pointing into the tensor have the -1 statistics.

```
1 >>> Q = gtn.einsum('ijk->kij',T)
2 >>> print('shape=',Q.shape,', stats=',Q.statistics)
3 shape= (4, 4, 8) , stats= (-1, 1, -1)
```

In this example, the third index is moved to the beginning without any contraction taking place. Note that swapping of any two fermionic indices introduces a sign factor to the coefficient matrix. This sign factor is automatically calculated by `grassmanntn.einsum()`.

Two or more tensors can be contracted:

```
1 >>> X = gtn.einsum('ijk, kia->ja',T,S)
2 >>> print('shape=',X.shape,', stats=',X.statistics)
3 shape= (8, 6) , stats= (-1, 0)
```

If the contraction is complete, the function returns a scalar:

```
1 >>> trM = gtn.einsum('ii',M)
2 >>> print(type(trM),trM)
3 <class 'numpy.float64'> 0.8348875099871086
```

The contraction can be done with more than 2 repeating indices if it is bosonic:

```
1 >>> R = gtn.einsum('kia,aa->ik',S,G)
2 >>> print('shape=',R.shape,', stats=',R.statistics)
3 shape= (4, 4) , stats= (-1, 1)
```

## 3.3 Tensor reshaping

Grassmann tensor can be reshaped similarly to the traditional multidimensional array. However, joining and splitting the tensor legs also introduce an additional sign factor to

the coefficients (see appendix A.3). To compute such a sign factor, the reshaping function must know the statistics of the target tensor. The following example shows how to reshape an order-4 tensor with the statistics (1,1,-1,-1) into an order-2 tensor with the statistics (1,-1)

```
>>> import grassmanntn as gtn
>>> A = gtn.random(shape=(4,4,4,4),statistics=(1,1,-1,-1))
>>> M = A.join_legs('(ij)(kl)',intermediate_stat=(1,-1))
```

In this example, the tensor `A` is reshaped with the function `join_legs()`. The first argument instructs how the tensor is reshaped; i.e., `(ij)(kl)` means that the first two indices `(ij)` are grouped into one index and similarly for `(kl)`. The statistics of the reshaped legs are specified by the argument `intermediate_stat`, which is `(1,-1)`. This means that the leg `(ij)` and `(kl)` has the `+1` and `-1` statistics, respectively. The dimensions of the reshaped legs are computed automatically. A diagrammatic representation of this reshaping process is shown in Figure 6.

Splitting the legs can be done in a similar way but with slightly different arguments. The following example shows how to reshape the order-2 tensor above back to the original order-4 tensor with the function `split_legs()`:

```
>>> A2 = M.split_legs('(ij)(kl)',intermediate_stat=(1,-1),
...                     final_stat=(1,1,-1,-1),final_shape=(4,4,4,4))
```

In this example, the first argument tells the function how the two legs should be split. Namely, the parent object `M` has two legs, so there must be two enclosed parentheses, which are `(ij)` and `(kl)`. In each parenthesis, the number of indices dictates how many legs it should be split into; i.e., both legs are split into two legs. The argument `final_stat` and `final_shape` tell the statistics and the shape of the reshaped tensor. The argument `intermediate_stat`, of which we will explain its significance below, should be the same as the parent object's statistics in most cases.

One can check that `A` and `A2` are the same by computing the norm of the difference:

```
>>> A2 = A2.force_encoder("canonical") # convert A2 to be in
>>>                                    # the same encoder as A
>>>                                    # to compute A-A2
>>> print((A-A2).norm) # is equal to zero if A=A2
0.0
```

Both `join_legs()` and `split_legs()` are designed to work in the most general cases where fermionic legs, conjugated legs, and bosonic legs, are simultaneously involved. In such cases, the argument `intermediate_stat` plays a crucial role. The joining process can be summarized in the following steps:

1. Consider the grouping $(I_1 \cdots I_m J_1 \cdots J_n i_1 \cdots i_p) \mapsto X$ where $I_a$, $J_b$, and $i_c$ are of `+1`, `-1`, and `0` statistics, respectively. $X$ is the joined leg.

2. The fermionic indices are first joined into a single fermionic index $K$ then bosonic indices are joined into a single bosonic index $k$:

$$(I_1 \cdots I_m J_1 \cdots J_n i_1 \cdots i_p) \mapsto (K i_1 \cdots i_p) \mapsto (Kk).$$

   If `intermediate_stat` of this grouping is `+1`, this fermion is non-conjugated. If it is `-1`, the intermediate fermion is conjugated. If there are only bosonic indices, `intermediate_stat` must be `0`. The statistic of the intermediate fermion affects the sign factor according to the prescription described in appendix A.3.

3. The user has the option to switch the coefficient format at this point (with the optional argument `make_format`; see the documentation for more details). Usually, this doesn't matter except when we want to perform matrix manipulation, where the coefficient must be in the matrix format.

4. Finally, the intermediate fermionic index $K$ and the bosonic indices $k$ are joined $(Kk) \mapsto X$ where $K$ is in the parity-preserving encoder and $X = K + d \times k$ (with $d$ being the dimension of the fermionic leg $K$).

Note that since $d$ must always be even, the parity of $X$ and $K$ are the same. This means that the Grassmann parity of the fermionic leg $K$ is preserved in $X$ even if $X$ contains bosonic degrees of freedom.

It should be stressed that the hybrid leg $X$ **does not** furnish a representation of the Grassmann algebra since all information of fermionic degrees of freedom (except its parity) is polluted by the bosonic degrees of freedom. Because of this, if the user wants to split the hybrid leg $X$ back to $(I_1 \cdots I_m J_1 \cdots J_n i_1 \cdots i_p)$, they have to specify not only `final_statistics` and `final_shape`, but also `intermediate_stat` of the intermediate index $K$ as well. In most cases, where bosonic indices are not involved, the intermediate statistics can be taken to be the same as the parent object's statistics.

An example of the case where the hybrid legs are created is when one wants to perform tensor decomposition of a hybrid tensor such as $\mathcal{T}_{\psi i \phi}$ into $\int_{\bar{\xi}\xi} \mathcal{A}_{\psi i \xi} \mathcal{B}_{\bar{\xi}\phi}$. In this case, the fermion $\psi$ and a bosonic index $i$ are necessarily joined into a hybrid leg first. After the decomposition, it is then split back into $\psi$ and $i$. This process, however, can be conveniently done by the functions `svd()` and `eig()` (see 3.4).

## 3.4   Tensor decomposition

Singular value decomposition (SVD) plays a central role in the low-rank approximation of various tensor network algorithms. The SVD can be generalized for Grassmann tensors (gSVD), which is formulated in Appendix A.7. Let

$$\mathcal{T}_{\psi_1\cdots\bar{\phi}_1\cdots i_1\cdots\psi'_1\cdots\bar{\phi}'_1\cdots k_1\cdots} = \sum_{\{I\},\{J\},\{K\},\{L\}} T_{I_1\cdots J_1\cdots i_1\cdots K_1\cdots L_1\cdots k_1}\psi_1^{I_1}\cdots\bar{\phi}_1^{J_1}\cdots\psi'^{K_1}_1\cdots\bar{\phi}'^{L_1}_1\cdots \quad (3.8)$$

be a general Grassmann tensor with indices of various statistics. Its gSVD of the form

$$\mathcal{T}_{\psi_1\cdots\bar{\phi}_1\cdots i_1\cdots\psi'_1\cdots\bar{\phi}'_1\cdots k_1\cdots} = \int_{\bar{\xi}\xi,\bar{\zeta}\zeta}\mathcal{U}_{\psi_1\cdots\bar{\phi}_1\cdots i_1\cdots\xi}\Sigma_{\bar{\xi}\zeta}\mathcal{V}_{\bar{\zeta}\psi'_1\cdots\bar{\phi}'_1\cdots k_1\cdots}, \quad (3.9)$$

where $\Sigma_{\bar{\xi}\sigma}$ is the diagonal singular value matrix, can be computed using `grassmanntn` with the command (an example with two indices of each type)

```
U, S, V = T.svd('I1 I2 J1 J2 i1 i2 | K1 K2 L1 L2 k1 k2')
```

In this example, the indices on the opposite sides of the renormalized legs are separated by the | indicator.

eigenvalue decomposition can also be done if the tensor is Hermitian (see appendix A.5 for definitions). In this case, the two unitary tensors $U$ and $V$ are conjugate to each other, and $\Sigma$ becomes the eigenvalue matrix.

**Examples**

Consider a three-legged tensor:

```
>>> import grassmanntn as gtn
>>> A   = gtn.random(shape=(4,4,4),statistics=(1,1,-1))
```

Its singular value decomposition with the renormalized leg between the first and the last two legs can be computed by

```
>>> U, S, V = A.svd('i|jk')
```

One can check if the decomposition is correct by reconstructing the original tensor and measuring the error:

```
>>> USV = gtn.einsum('ia,ab,bjk->ijk',U,S,V)
>>> print( (A-USV).norm ) # is equal to zero if A=USV
7.557702638948695e-16
```

To demonstrate the eigenvalue decomposition, consider a Hermitian tensor

```
>>> H = gtn.einsum('jki,iJK->jkJK',A.hconjugate('i|jk'),A)
```

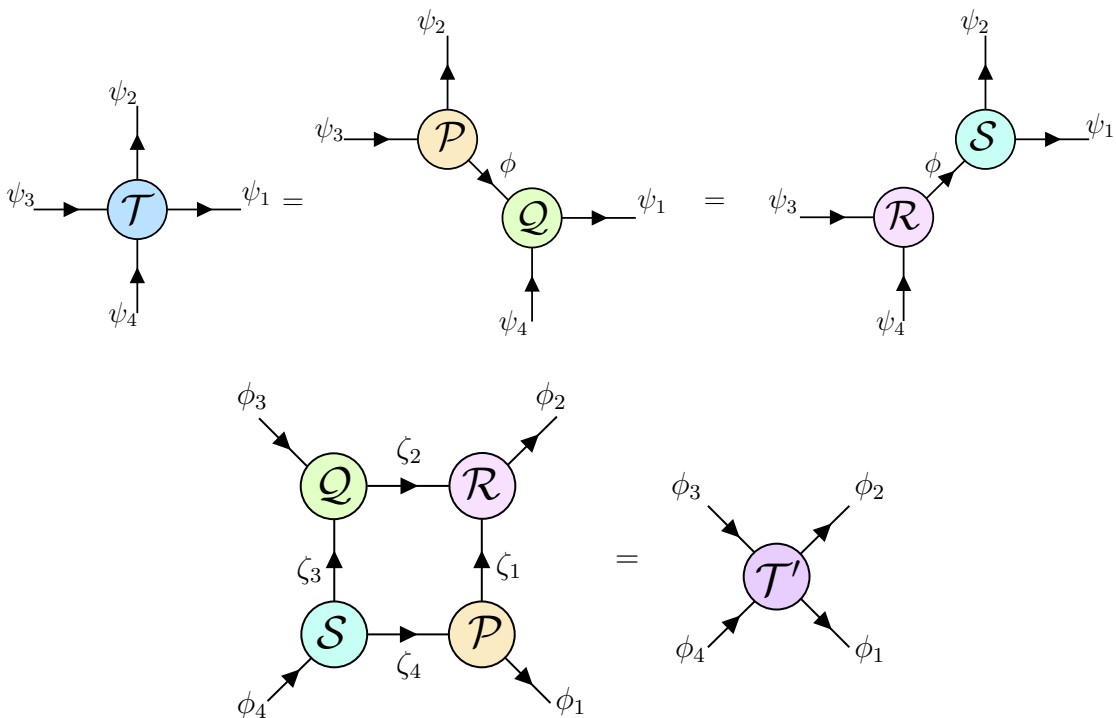

Figure 7: (Top) The two configurations of Grassmann SVD. (Bottom) The construction of the coarse-grained tensor.

Here, we form a Hermitian tensor `H` by contracting `A` with its Hermitian conjugate. Now we can compute the eigenvalue decomposition

```
1 >>> U, S, V = H.eig('jk|JK')
2 >>> USV = gtn.einsum('jka,ab,bJK->jkJK',U,S,V)
3 >>> print( (H-USV).norm ) # is equal to zero if H=USV
4 9.177321373036202e-15
```

We can also show that `U` and `V` are conjugate to each other:

```
1 >>> print((U-V.hconjugate('a|JK')).norm) # is zero if U is V-conjugate
2 0.0
```

# 4 Coding examples

## 4.1 Levin-Nave TRG

The initial version of the tensor renormalization group algorithms was developed to handle the Ising model [11], which is a two-dimensional spin system. In their approach, a coarse-graining procedure is utilized to perform a scale transformation, akin to the conventional

real-space renormalization group transformation. This can be directly generalized to the Grassmann tensor network, which has been demonstrated with the Schwinger model [29, 24, 30], among others.

The Grassmann TRG method assumes that the lattice is periodic with an order-4 tensor

$$\mathcal{T}_{\psi_1\psi_2\bar{\psi}_3\bar{\psi}_4}. \tag{4.1}$$

The tensor is periodic in the $x$ (1 and 3) axis and $y$ (2 and 4) axis. At the even and odd sites, the tensor is decomposed with different configurations of SVD:

$$\mathcal{T}_{\psi_1\psi_2\bar{\psi}_3\bar{\psi}_4} = \int_{\bar{\zeta}\zeta}\int_{\bar{\xi}\xi} \mathcal{U}^{\mathrm{E}}_{\psi_2\bar{\psi}_3\zeta}\Sigma^{\mathrm{E}}_{\bar{\zeta}\xi}\mathcal{V}^{\mathrm{E}}_{\bar{\xi}\bar{\psi}_4\psi_1} = \int_{\bar{\phi}\phi}\mathcal{P}_{\psi_2\bar{\psi}_3\phi}\mathcal{Q}_{\bar{\phi}\bar{\psi}_4\psi_1} \qquad \text{(even sites)} \tag{4.2}$$

$$= \int_{\bar{\zeta}\zeta}\int_{\bar{\xi}\xi} \mathcal{U}^{\mathrm{O}}_{\bar{\psi}_3\bar{\psi}_4\zeta}\Sigma^{\mathrm{O}}_{\bar{\zeta}\xi}\mathcal{V}^{\mathrm{O}}_{\bar{\xi}\psi_1\psi_2} = \int_{\bar{\phi}\phi}\mathcal{R}_{\bar{\psi}_3\bar{\psi}_4\phi}\mathcal{S}_{\bar{\phi}\psi_1\psi_2} \qquad \text{(odd sites)}. \tag{4.3}$$

Here, both $\mathcal{P}$ and $\mathcal{Q}$ absorb a square root of $\Sigma^{\mathrm{E}}$ (and similarly for $\mathcal{R}$, $\mathcal{S}$, and $\Sigma^{\mathrm{O}}$), where the square root of a diagonal tensor is defined by

$$\Sigma_{\bar{\zeta}\xi} = \sum_I \lambda_I \sigma_I \bar{\zeta}^I \xi^I \quad \to \quad \sqrt{\Sigma}_{\bar{\zeta}\xi} = \sum_I \sqrt{\lambda_I} \sigma_I \bar{\zeta}^I \xi^I. \tag{4.4}$$

The coarse-grained tensor can then be constructed via

$$\mathcal{T}'_{\phi_1\phi_2\bar{\phi}_3\bar{\phi}_4} = \int_{\substack{\bar{\zeta}_1\zeta_1,\bar{\zeta}_2\zeta_2,\\ \bar{\zeta}_3\zeta_3,\bar{\zeta}_4\zeta_4}} \mathcal{S}_{\bar{\phi}_4\zeta_4\zeta_3}\mathcal{Q}_{\bar{\phi}_3\bar{\zeta}_3\zeta_2}\mathcal{P}_{\zeta_1\bar{\zeta}_4\phi_1}\mathcal{R}_{\bar{\zeta}_2\bar{\zeta}_1\phi_2}. \tag{4.5}$$

This procedure can be computed with the following function:

```
>>> import grassmanntn as gtn
>>> def LevinNaveTRG(T):
...     # Input a site tensor T
...     # Return a renormalized tensor Tprime
...     TE = gtn.einsum('i1 i2 i3 i4 -> i2 i3 i4 i1',T) # Even arrangement
...     TO = gtn.einsum('i1 i2 i3 i4 -> i3 i4 i1 i2',T) # Odd arrangement
...     UE, SE, VE = TE.svd('i2 i3 | i4 i1') #Even-site SVD
...     UO, SO, VO = TO.svd('i3 i4 | i1 i2') #Odd-site SVD
...     sqSE = gtn.sqrt(SE) # the square-root of the singular value
...     sqSO = gtn.sqrt(SO) # the square-root of the singular value
...     P = gtn.einsum('i2 i3 a, ab -> i2 i3 b', UE,sqSE)
...     Q = gtn.einsum('ab, b i4 i1 -> a i4 i1', sqSE,VE)
...     R = gtn.einsum('i3 i4 a, ab -> i3 i4 b', UO,sqSO)
...     S = gtn.einsum('ab, b i1 i2 -> a i1 i2', sqSO,VO)
...     SQ = gtn.einsum('i4 j4 j3, i3 j3 j2 -> i3 i4 j2 j4',S,Q) # S*Q
...     PR = gtn.einsum('j1 j4 i1, j2 j1 i2 -> j4 j2 i1 i2',P,R) # P*R
```

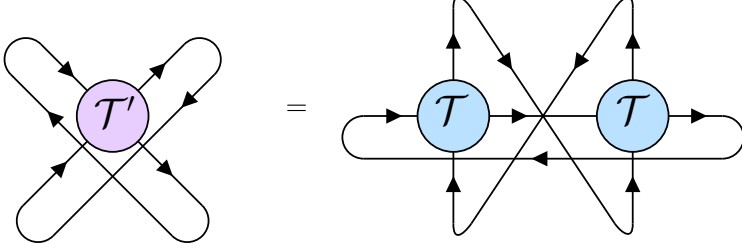

Figure 8: The equivalence of the tensor trace before and after performing the coarse-graining procedure.

```
17 ...      Tprime = gtn.einsum('i3 i4 j2 j4,j4 j2 i1 i2->i1 i2 i3 i4',SQ,PR)
18 ...      return Tprime
19 ...
20 >>> T = gtn.random(shape=(4,4,4,4),statistics=(1,1,-1,-1))
21 >>> Tprime = LevinNaveTRG(T) # Performing TRG of a random tensor
22 >>> Tprime.info("Tprime")
23
24          name: Tprime
25    array type: dense
26         shape: (16, 16, 16, 16)
27       density: 32768 / 65536 ~ 50.0 %
28    statistics: (1, 1, -1, -1)
29        format: standard
30       encoder: canonical
31        memory: 512.2 KiB
32          norm: 21.0229552947218
```

To test if our result is correct, one way is to compute the trace directly and via the TRG. If our TRG algorithm is correct, the following relation should hold:

$$\int_{\bar{\phi}_1\phi_1,\bar{\phi}_2\phi_2} \mathcal{T}'_{\phi_1\phi_2\bar{\phi}_1\bar{\phi}_2} = \int_{\substack{\bar{\psi}_1\psi_1,\bar{\psi}_2\psi_2,\\ \bar{\psi}_3\psi_3,\bar{\psi}_4\psi_4}} \mathcal{T}_{\psi_1\psi_2\bar{\psi}_3\bar{\psi}_4}\mathcal{T}_{\psi_3\psi_4\bar{\psi}_1\bar{\psi}_2}. \tag{4.6}$$

This equivalence is depicted diagrammatically in figure 8. The two traces can be shown to be indeed the same:

```
1 >>> trace1 = gtn.einsum('i1 i2 i3 i4, i3 i4 i1 i2',T,T)
2 >>> trace2 = gtn.einsum('i1 i2 i1 i2',Tprime)
3 >>> print('trTT    =', trace1,'\ntrTprime =', trace2)
4 trTT     = -0.20488002067705247
5 trTprime = -0.20488002067708644
```

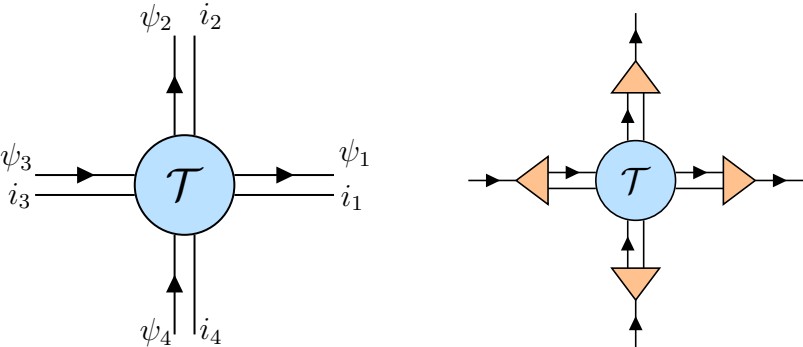

Figure 9: The example tensor (4.7) (left) and how the isometries (triangles) are applied (right).

## 4.2   Isometry tensor computation

A standard operation in tensor renormalization group algorithms is the computation of the isometry or the squeezer of a given set of tensor legs [44, 18, 13]. Consider the following Grassmann tensor

$$\mathcal{T}_{\psi_1\psi_2\bar\psi_3\bar\psi_4 i_1 i_2 i_3 i_4} = \sum_{I_1,I_2,I_3,I_4} T_{I_1 I_2 I_3 I_4 i_1 i_2 i_3 i_4} \psi_1^{I_1} \psi_2^{I_2} \bar\psi_3^{I_3} \bar\psi_4^{I_4}, \tag{4.7}$$

where $\psi_a$ are 2-bit fermions and $i_a$ are bosonic indices with dimension 3. The tensor is assumed to be periodic in the $x$ (1 and 3) axis and $y$ (2 and 4) axis. This is depicted as a diagram in figure 9 (left).

Let us set up this tensor with `grassmanntn.random()`

```
1 >>> import grassmanntn as gtn
2 >>> T = gtn.random(shape=(4,4,4,4,3,3,3,3),
3 ...                         statistics=(1,1,-1,-1,0,0,0,0))
4 >>> T.info("Before truncation")
5
6           name: Before truncation
7     array type: dense
8          shape: (4, 4, 4, 4, 3, 3, 3, 3)
9        density: 10368 / 20736 ~ 50.0 %
10    statistics: (1, 1, -1, -1, 0, 0, 0, 0)
11         format: standard
12        encoder: canonical
13         memory: 162.3 KiB
14           norm: 58.636872692182166
15
16 >>> trT = gtn.einsum("IJIJijij",T)
```

```
17  >>> print("original trace:",trT)
18  original trace: -25.907152855719076
```

To squeeze the legs, we have to rearrange the indices so that the legs to be squeezed are separated from the others with `grassmanntn.einsum()`. For future convenience, the non-conjugated legs (1 and 2 directions) will be separated to the right while the conjugated legs (3 and 4 directions) will be separated to the left.

```
1  >>> T1 = gtn.einsum("IJKLijkl -> JKLjkl Ii",T)
2  >>> T2 = gtn.einsum("IJKLijkl -> IKLikl Jj",T)
3  >>> T3 = gtn.einsum("IJKLijkl -> Kk IJLijl",T)
4  >>> T4 = gtn.einsum("IJKLijkl -> Ll IJKijk",T)
```

We next perform the Hermitian conjugation.

```
1  >>> cT1 = T1.hconjugate("JKLjkl|Ii")
2  >>> cT2 = T2.hconjugate("IKLikl|Jj")
3  >>> cT3 = T3.hconjugate("Kk|IJLijl")
4  >>> cT4 = T4.hconjugate("Ll|IJKijk")
```

The Hermitian tensor can then be formed by summing out the 'environment' indices.

```
1  >>> M1 = gtn.einsum('Xx JKLjkl,JKLjkl Yy -> Xx Yy',cT1,T1)
2  >>> M2 = gtn.einsum('Xx IKLikl,IKLikl Yy -> Xx Yy',cT2,T2)
3  >>> M3 = gtn.einsum('Xx IJLijl,IJLijl Yy -> Xx Yy',T3,cT3)
4  >>> M4 = gtn.einsum('Xx IJKijk,IJKijk Yy -> Xx Yy',T4,cT4)
```

We next perform the eigenvalue decomposition and obtain the entanglement entropy of each case

```
1   >>> #eigenvalue decomposition
2   >>> U1,S1,V1 = M1.eig("Xx|Yy")
3   >>> U2,S2,V2 = M2.eig("Xx|Yy")
4   >>> U3,S3,V3 = M3.eig("Xx|Yy")
5   >>> U4,S4,V4 = M4.eig("Xx|Yy")
6   >>> #Get the entanglement spectrum
7   >>> import numpy as np
8   >>> Spect1 = np.diag(S1.force_format("matrix").data)
9   >>> Spect2 = np.diag(S2.force_format("matrix").data)
10  >>> Spect3 = np.diag(S3.force_format("matrix").data)
11  >>> Spect4 = np.diag(S4.force_format("matrix").data)
12  >>> #Compute the entanglement entropy
13  >>> Ent1 = sum([ -s*np.log(s+1e-16) for s in Spect1 ])
14  >>> Ent2 = sum([ -s*np.log(s+1e-16) for s in Spect2 ])
15  >>> Ent3 = sum([ -s*np.log(s+1e-16) for s in Spect3 ])
16  >>> Ent4 = sum([ -s*np.log(s+1e-16) for s in Spect4 ])
```

In each direction, we pick the unitary matrix with a smaller entanglement entropy as the isometry.

```
1 >>> # Get the isometries
2 >>> Ux = U1 if Ent1<Ent3 else U3
3 >>> Uy = U2 if Ent2<Ent4 else U4
4 >>> cUx = Ux.hconjugate("Xx|A")
5 >>> cUy = Uy.hconjugate("Xx|A")
```

And finally, we apply these isometries on the original tensor's legs.

```
1 >>> # Apply the isometries to the tensor
2 >>> Tprime = gtn.einsum('IJKLijkl,IiA -> AJKLjkl',T,Ux)
3 >>> Tprime = gtn.einsum('AJKLjkl ,JjB -> ABKLkl ',Tprime,Uy)
4 >>> Tprime = gtn.einsum('ABKLkl  ,CKk -> ABCLl  ',Tprime,cUx)
5 >>> Tprime = gtn.einsum('ABCLl   ,DLl -> ABCD   ',Tprime,cUy)
6 >>> Tprime.info("After truncation")
7
8         name: After truncation
9   array type: dense
10        shape: (16, 16, 16, 16)
11      density: 10368 / 65536 ~ 15.8203125 %
12   statistics: (1, 1, -1, -1)
13       format: standard
14      encoder: canonical
15       memory: 512.2 KiB
16         norm: 58.63687269218216
17
18 >>> trTprime = gtn.einsum("IJIJ",Tprime)
19 >>> print("truncated tensor trace:",trTprime)
20 truncated tensor trace: -25.907152855719076
```

The isometries in this example merge a fermionic leg and a bosonic leg into a new fermionic leg. This new leg is a proper representation of the Grassmann algebra, so it can be treated as a regular fermionic leg. Note how the tensor trace is not affected by the isometry.

# 5   Summary

In this paper, we introduce `grassmanntn`, a Python package designed to simplify the coding of Grassmann tensor network computation. The Grassmann tensor network is a useful tool for handling a large fermionic system, but the sign factor which is an inherent nature of Grassmann numbers makes the coding difficult and prone to mistakes. To that end,

`grassmanntn` computes the sign factor automatically. With the declarative programming approach, most of the functions are designed to work with the tensors' subscripts as the input. As such, the code can be easily translated from the symbolic expression. Two use case examples are given: the Levin-Nave TRG algorithm and the computation of isometries. Additionally, the package has also been recently used for the $N_f$-flavor gauge theory [13].

While the current version of `grassmanntn` can be successfully used in realistic computations, there is still more room for improvement. In particular, we plan to optimize the function `einsum` which has a bottleneck in the operational time in the sign factor tensor computation. In that aspect, path optimization will clearly help improve the speed. Another future plan is the implementation of basic Grassmann arithmetic, which can be used to construct the initial tensor from a given action without the help of external tools.

We encourage the community to use and test `grassmanntn` and give us feedback so that we can improve the package further. We hope that `grassmanntn` will become a tool that makes the Grassmann tensor network more accessible to new researchers and makes theoretical developments in both high energy and condensed matter physics.

# Acknowledgments

We would like to thank Jun Nishimura and Kouichi Okunishi for their valuable discussions. This work is supported by a Grant-in-Aid for Transformative Research Areas "The Natural Laws of Extreme Universe—A New Paradigm for Spacetime and Matter from Quantum Information" (KAKENHI Grant No. JP21H05191) from JSPS of Japan.

# A    Formulation

In this section, we formulate the concept of Grassmann tensor in the bottom-up approach. The main result of this formulation is that we have systematically introduced the process of joining and splitting the fermionic legs in the general case where the legs can be either conjugated or non-conjugated. Furthermore, we also re-introduced concepts such as Hermitian conjugation and matrix decomposition in a way that can be clearly and directly related to the non-Grassmann counterparts.

## A.1    Grassmann algebra

Given an $n$-dimensional vector space $V = \mathrm{span}(\theta_1, \cdots, \theta_n)$, a *Grassmann algebra* $\Lambda(V)$ is defined as an algebra of Grassmann generators $\theta_1, \cdots, \theta_n$ and their exterior (anticommu-

tative) products

$$\Lambda(V) \equiv \mathbb{C} \oplus V \oplus (V \wedge V) \oplus \cdots \oplus \underbrace{(V \wedge V \wedge \cdots \wedge V)}_{n}. \tag{A.1}$$

Elements of the Grassmann algebra are called the *Grassmann numbers* [45]. Essentially, Grassmann algebra describes a system of numbers $\theta_1, \cdots, \theta_n$ with the rule that they are anticommuting with each other: $\theta_a \theta_b = -\theta_b \theta_a$ for $a, b = 1, \cdots, n$.

If a Grassmann number is commuting/anti-commuting with all generators, we say that it has an even/odd Grassmann parity. In general, the commutativity of two Grassmann numbers $x_1, x_2 \in \Lambda(V)$ is given by

$$x_1 x_2 = (-)^{p(x_1)p(x_2)} x_2 x_1 \tag{A.2}$$

where $p(x) = 0$ if $x$ is Grassmann even and $p(x) = 1$ if it is Grassmann odd. It follows straightforwardly that a square of any Grassmann-odd numbers always vanishes. Consequently, any element $\mathcal{A} \in \Lambda(V)$ can be written uniquely by the sum

$$\mathcal{A} = \sum_{i_1, \cdots, i_n \in \{0,1\}} A_{i_1 \cdots i_n} \theta_1^{i_1} \cdots \theta_n^{i_n}, \tag{A.3}$$

for some $A_{i_1 \cdots i_n} \in \mathbb{C}$. In contrast to the polynomial expansion of complex numbers, where the power must be truncated at some large number, $i_a$ is already truncated at 1 because of the property that the square (or higher power) of $\theta_a$ identically vanishes.

The integral of a Grassmann number, known as the Berezin integral, can be defined as follows [46]:

$$\int d\theta \theta \equiv 1, \tag{A.4}$$

$$\int d\theta 1 \equiv 0. \tag{A.5}$$

Keep in mind that these operators are also anticommuting. It is easy to show that

$$\int d\theta' d\theta e^{-\theta'\theta} \theta^i \theta'^j = \delta_{ij}, \tag{A.6}$$

for any two generators $\theta$ and $\theta'$. This identity will become important when we discuss tensor contraction below.

A Grassmann algebra $\mathfrak{v} = \Lambda(V)$ of an $n$-dimensional vector space $V$ is itself a vector space with $\dim(\mathfrak{v}) = 2^n$. Basis vectors of $\mathfrak{v}$ can be indexed by a parameter $I$, which we will

call the *composite index*. Specifically, a basis vector $\psi^I$ of $\mathfrak{v}$ with $I = (i_1, \cdots, i_n)$ is defined by

$$\psi^I \equiv \theta_1^{i_1} \cdots \theta_n^{i_n}. \tag{A.7}$$

Any Grassmann number $\mathcal{A} \in \mathfrak{v}$ can then be written as (see (A.3) for comparison)

$$\mathcal{A} = \sum_I A_I \psi^I. \tag{A.8}$$

We will refer to the symbol $\psi = (\theta_1, \cdots, \theta_n)$ as an *n*-bit *fermionic index*, or just a *fermion*. Note that $\psi$ may also be referred to as the 'multi-component Grassmann number' in the literature. We can define Grassmann parity of $\psi^I$ by

$$p(I) = \sum_{a=1}^n i_a, \tag{A.9}$$

which is the sum of the occupation number of the generators. Similar to (A.2), the commutativity of two Grassmann numbers are given by

$$\psi_1^{I_1} \psi_2^{I_2} = (-)^{p(I_1)p(I_2)} \psi_2^{I_2} \psi_1^{I_1} \tag{A.10}$$

## A.2   Dual algebra and Grassmann contraction

For every Grassmann algebra $\mathfrak{v} = \Lambda(V)$, there is a dual Grassmann algebra $\bar{\mathfrak{v}} = \Lambda(\bar{V})$ with $\bar{V} = \operatorname{span}(\bar{\theta}_1, \cdots, \bar{\theta}_n)$. The generators $\bar{\theta}_a$ and $\theta_a$ are said to be *dual* or *conjugated* to each other. The dual Grassmann algebra $\bar{\mathfrak{v}}$ can also be defined as a set of all linear maps that maps any element of $\mathfrak{v}$ into a complex scalar via an operation called *Grassmann contraction*. A Grassmann contraction between a Grassmann vector $\mathcal{A}_\psi = \sum_I A_I \psi^I \in \mathfrak{v}$ and a dual vector $\mathcal{B}_{\bar{\psi}} = \sum_J B_J \bar{\psi}^J \in \bar{\mathfrak{v}}$ is defined by

$$\int_{\bar{\psi}\psi} \mathcal{A}_\psi \mathcal{B}_{\bar{\psi}} \equiv \int \left( \prod_{a=1}^n d\bar{\theta}_a d\theta_a e^{-\bar{\theta}_a \theta_a} \right) \mathcal{A}_\psi \mathcal{B}_{\bar{\psi}} \tag{A.11}$$

$$= \sum_{\{i\},\{j\}} \int \left( \prod_{a=1}^n d\bar{\theta}_a d\theta_a e^{-\bar{\theta}_a \theta_a} \right) \left( A_{i_1 \cdots i_n} \theta_1^{i_1} \cdots \theta_n^{i_n} \right) \left( B_{j_1 \cdots j_n} \bar{\theta}_1^{j_1} \cdots \bar{\theta}_n^{j_n} \right)$$

$$= \sum_{\{i\},\{j\}} \int \left( \prod_{a=1}^n d\bar{\theta}_a d\theta_a e^{-\bar{\theta}_a \theta_a} \theta_a^{i_a} \bar{\theta}_a^{j_a} \right) \left( \prod_{a>b} (-)^{i_a j_b} \right) A_{i_1 \cdots i_n} B_{j_1 \cdots j_n}$$

$$= \sum_{\{i\},\{j\}} \left( \prod_{a=1}^n \delta_{i_a j_a} \right) \left( \prod_{a>b} (-)^{i_a j_b} \right) A_{i_1 \cdots i_n} B_{j_1 \cdots j_n}$$

$$= \sum_{\{i\}} \left( \prod_{a>b} (-)^{i_a i_b} \right) A_{i_1 \cdots i_n} B_{i_1 \cdots i_n}, \tag{A.12}$$

which is a complex number. In the equations above, the product symbol $\prod_a$ is ordered in a way that terms with smaller $a$ are to the left of those with larger $a$. The sign factor

$$\sigma_I = \prod_{a>b} (-)^{i_a i_b} \tag{A.13}$$

comes from rearranging the Grassmann number from the second line to the third line. Using (A.12), it is easy to derive the orthogonality relation

$$\int_{\bar{\psi}\psi} \psi^I \bar{\psi}^J = \delta_{IJ} \sigma_I. \tag{A.14}$$

The contraction (A.12) can then be rewritten in terms of composite indices as

$$\int_{\bar{\psi}\psi} \mathcal{A}_\psi \mathcal{B}_{\bar{\psi}} = \sum_I \sigma_I A_I B_I. \tag{A.15}$$

## A.3  Joining and splitting algebras

In the tensor network computation, multiple tensor legs sometimes need to be merged into a single leg. In our context, this corresponds to the joining of the Grassmann algebra of each leg into a single Grassmann algebra: $(\mathfrak{v}_1, \cdots, \mathfrak{v}_m, \bar{\mathfrak{u}}_1, \cdots, \bar{\mathfrak{u}}_n) \mapsto \mathfrak{w}$. This can be done in two steps: 1) the algebras are first combined with the direct sum $\mathfrak{t}$; 2) The joined algebra $\mathfrak{w}$ is then formed as a graded tensor product of $\mathfrak{t}$:

$$\mathfrak{w} \equiv \mathbb{C} \oplus \mathfrak{t} \oplus (\mathfrak{t} \otimes \mathfrak{t}) \oplus \cdots \oplus \underbrace{(\mathfrak{t} \otimes \mathfrak{t} \otimes \cdots \mathfrak{t})}_{m+n}, \tag{A.16}$$

$$\bar{\mathfrak{w}} \equiv \mathbb{C} \oplus \bar{\mathfrak{t}} \oplus (\bar{\mathfrak{t}} \otimes \bar{\mathfrak{t}}) \oplus \cdots \oplus \underbrace{(\bar{\mathfrak{t}} \otimes \bar{\mathfrak{t}} \otimes \cdots \bar{\mathfrak{t}})}_{m+n}; \tag{A.17}$$

$$\mathfrak{t} = \mathfrak{v}_1 \oplus \cdots \oplus \mathfrak{v}_m \oplus \bar{\mathfrak{u}}_1 \oplus \cdots \oplus \bar{\mathfrak{u}}_n, \tag{A.18}$$

$$\bar{\mathfrak{t}} = \bar{\mathfrak{v}}_1 \oplus \cdots \oplus \bar{\mathfrak{v}}_m \oplus \mathfrak{u}_1 \oplus \cdots \oplus \mathfrak{u}_n. \tag{A.19}$$

Component-wise, the fermions $\psi_a^{I_a} \in \mathfrak{v}_a$ and $\phi_b^{J_b} \in \mathfrak{u}_b$ are joined into $\xi^K \in \mathfrak{w}$ with the following prescription:

$$\xi^K \equiv \psi_1^{I_1} \cdots \psi_m^{I_m} \bar{\phi}_1^{J_1} \cdots \bar{\phi}_n^{J_n}, \tag{A.20}$$

$$\bar{\xi}^K \equiv \bar{\psi}_1^{I_1} \cdots \bar{\psi}_m^{I_m} \phi_1^{J_1} \cdots \phi_n^{J_n} \times \prod_{b=1}^n (-)^{p(J_b)}, \tag{A.21}$$

with the contraction defined by

$$\int_{\bar\xi\xi} \equiv \int_{\bar\psi_1\psi_1} \cdots \int_{\bar\psi_m\psi_m} \int_{\bar\phi_1\phi_1} \cdots \int_{\bar\phi_n\phi_n} . \tag{A.22}$$

Note how $\xi^K$ and $\bar\xi^K$ are defined differently, with the conjugated one having an extra sign factor. This is to ensure that the composite algebras $\mathfrak{w}$ and $\bar{\mathfrak{w}}$ are dual to each other in the sense that $\xi^K$ and $\bar\xi^K$ are contracted by $\int_{\bar\xi\xi}$ with the identity (A.14)[4]. Splitting the algebras can also be done in reverse order of the joining process.

## A.4   Grassmann tensors

A *Grassmann tensor algebra* $\mathscr{T}$ is defined to be a graded tensor product of several Grassmann algebras $\mathfrak{v}_a$ and dual algebras $\bar{\mathfrak{u}}_b$:

$$\mathscr{T} \equiv \mathbb{C} \oplus \mathfrak{t} \oplus (\mathfrak{t} \otimes \mathfrak{t}) \oplus \cdots \oplus \underbrace{(\mathfrak{t} \otimes \mathfrak{t} \otimes \cdots \mathfrak{t})}_{m+n}; \tag{A.23}$$

$$\mathfrak{t} = \mathfrak{v}_1 \oplus \cdots \oplus \mathfrak{v}_m \oplus \bar{\mathfrak{u}}_1 \oplus \cdots \oplus \bar{\mathfrak{u}}_n. \tag{A.24}$$

If $\mathscr{T}$ is composed of $m$ Grassmann algebras and $n$ dual algebras, we say that the element of $\mathscr{T}$ is a *Grassmann tensor* of order $(m, n)$. If we treat $\mathscr{T}$ as a Grassmann algebra, $\mathscr{T}$ will be equivalent to the composite Grassmann algebra $\mathfrak{w}$ introduced in section A.3. The difference is that we still keep the fermions separated in this case. A Grassmann tensor $\mathcal{T} \in \mathscr{T}$ can always be represented by the sum

$$\mathcal{T}_{\psi_1\cdots\psi_m\bar\phi_1\cdots\bar\phi_n} = \sum_{I_1,\cdots,I_m,J_1,\cdots,J_n} T_{I_1\cdots I_m J_1\cdots J_n} \psi_1^{I_1} \cdots \psi_m^{I_m} \bar\phi_1^{J_1} \cdots \bar\phi_n^{J_n} \tag{A.25}$$

where $\psi_a^{I_a} \in \mathfrak{v}_a$, $\bar\phi_b^{I_b} \in \bar{\mathfrak{u}}_b$, and $T_{I_1\cdots J_n} \in \mathbb{C}$. A Grassmann tensor is called a *Grassmann vector* if it has one index (such as $\mathcal{A}_\psi$ or $\mathcal{B}_{\bar\psi}$). If it has one non-conjugated index and one conjugated index (such as $\mathcal{M}_{\bar\psi\phi}$), we call it a *Grassmann matrix*.

Two Grassmann tensors can be contracted if the contracted indices are dual to each other. The dual indices must be moved adjacent to each other first before we can perform the contraction. This introduces some sign factors in the coefficient tensor. The following example shows the contraction of the pair $(\phi, \bar\phi)$ between $\mathcal{A}_{\psi_1\phi\bar\psi_3}$ and $\mathcal{B}_{\bar\psi_2\bar\phi}$:

$$\mathcal{C}_{\psi_1\bar\psi_2\bar\psi_3} = \int_{\bar\phi\phi} \mathcal{A}_{\psi_1\phi\bar\psi_3} \mathcal{B}_{\bar\psi_2\bar\phi}$$

---

[4]Alternatively, the sign factor can be absorbed in the definition of $\int_{\bar\xi\xi}$. But this is not preferable since this makes the definition of the integral depend on its integrand.

$$
= \sum_{I_1, I_2, I_3, K, K'} A_{I_1 K I_3} B_{I_2 K'} \int_{\bar{\phi}\phi} \psi_1^{I_1} \phi^K \bar{\psi}_3^{I_3} \psi_2^{I_2} \bar{\phi}^{K'}
$$

$$
= \sum_{I_1, I_2, I_3, K, K'} A_{I_1 K I_3} B_{I_2 K'} (-)^{p(K')(p(I_2)+p(I_3))} \psi_1^{I_1} \left( \int_{\bar{\phi}\phi} \phi^K \bar{\phi}^{K'} \right) \bar{\psi}_3^{I_3} \bar{\psi}_2^{I_2}
$$

$$
= \sum_{I_1, I_2, I_3} \underbrace{\sum_K A_{I_1 K I_3} B_{I_2 K} (-)^{p(K)(p(I_2)+p(I_3))+p(I_2)p(I_3)} \sigma_K \psi_1^{I_1} \bar{\psi}_2^{I_2} \bar{\psi}_3^{I_3}}_{= C_{I_1 I_2 I_3}}. \tag{A.26}
$$

Keep in mind that the conjugated fermion must be on the right-hand side of the non-conjugated fermion in the formula (A.14). Also note that the contraction operator $\int_{\bar{\psi}\psi}$ is Grassmann-even, so it can be moved anywhere without introducing extra sign factors.

Grassmann tensors can be depicted diagrammatically similarly to the usual tensors. However, the conjugated and non-conjugated legs must be clearly distinguished. Following the convention given in Ref. [14], non-conjugated legs have an arrow pointing *away* from the tensor while conjugated legs have an arrow pointing *into* the tensor. For example, the diagram of (A.26) is given in figure 3.

## A.5   Unitary space

Unitary space is a vector space equipped with 1) an *inner product* and 2) a *Hermitian conjugation map* that maps between the vector space and its dual. In our context, the vector space refers to the order-1 tensor algebra (the space of Grassmann vectors) while the inner product is defined by

$$
\langle \mathcal{B}, \mathcal{A} \rangle \equiv \int_{\bar{\psi}\psi} \mathcal{B}_\psi^\dagger \mathcal{A}_{\bar{\psi}}. \tag{A.27}
$$

The conjugation map is defined on a vector and a matrix by

$$
\mathcal{A}_\psi = \sum_I A_I \psi^I \ \longrightarrow \ \mathcal{A}_{\bar{\psi}}^\dagger \equiv \sum_I A_I^* \sigma_I \bar{\psi}^I, \tag{A.28}
$$

$$
\mathcal{B}_{\bar{\psi}} = \sum_I B_I \bar{\psi}^I \ \longrightarrow \ \mathcal{B}_\psi^\dagger \equiv \sum_I B_I^* \sigma_I \psi^I, \tag{A.29}
$$

$$
\mathcal{M}_{\bar{\psi}\phi} = \sum_{I,J} M_{IJ} \bar{\psi}^I \phi^J \ \longrightarrow \ \mathcal{M}_{\bar{\phi}\psi}^\dagger \equiv \sum_{I,J} M_{IJ}^* \sigma_I \sigma_J \bar{\phi}^J \psi^I. \tag{A.30}
$$

The symbol $(\,\cdot\,)^*$ denotes complex conjugation. It is easy to see that performing the Hermitian conjugation twice gives the original object. An inner product of a Grassmann vector with itself is positive semi-definite:

$$
\langle \mathcal{A}, \mathcal{A} \rangle = \int_{\bar{\psi}\psi} \mathcal{A}_\psi^\dagger \mathcal{A}_{\bar{\psi}} = \sum_I |A_I|^2. \tag{A.31}
$$

For general tensors, conjugation can be done by joining the indices into two groups first (turning into a matrix), performing the conjugation, and finally splitting the indices. For example, considering

$$\mathcal{T}_{\psi_1\psi_2\psi_3\psi_4} = \sum_{I_1I_2I_3I_4} T_{I_1I_2I_3I_4}\psi_1^{I_1}\psi_2^{I_2}\psi_3^{I_3}\psi_4^{I_4}, \tag{A.32}$$

the conjugated with respect to the grouping $(\psi_1\psi_2)(\psi_3\psi_4)$ is given by

$$\mathcal{T}^{\dagger}_{(\bar{\psi}_3\bar{\psi}_4)(\bar{\psi}_1\bar{\psi}_2)} = \sum_{I_1I_2I_3I_4} T^*_{I_1I_2I_3I_4}\sigma_{(I_1,I_2)}\sigma_{(I_3,I_4)}(-)^{p(I_1)+p(I_2)+p(I_3)+p(I_4)}\bar{\psi}_3^{I_3}\bar{\psi}_4^{I_4}\bar{\psi}_1^{I_1}\bar{\psi}_2^{I_2}. \tag{A.33}$$

In the equation above,

$$\sigma_{(I_a,I_b)} = \sigma_{I_a}\sigma_{I_b}(-)^{p(I_a)p(I_b)} \tag{A.34}$$

is the sign factor (A.13) with the argument being the composite index $I = (I_a, I_b)$ and $(-)^{p(I_1)+p(I_2)+p(I_3)+p(I_4)}$ is the sign factor arising from index joining and splitting. It should be noted that performing conjugation with different index groupings gives a different result.

A Grassmann matrix is said to be *Hermitian* if $\mathcal{H}^{\dagger}_{\bar{\psi}\phi} = \mathcal{H}_{\bar{\psi}\phi}$. In other words, its coefficient tensor must satisfy the condition

$$H_{JI} = H^*_{IJ}\sigma_I\sigma_J. \tag{A.35}$$

The coefficient matrix of a Hermitian Grassmann matrix is **not** a Hermitian matrix. This peculiar statement will be clarified when we discuss the coefficient formats in section A.6. Although the coefficient of a Hermitian Grassmann matrix is seemingly counter-intuitive, one can check that it has all the right properties. For example, we can show that the eigenvalues of a Hermitian Grassmann matrix are all real by showing that its expectation value is always real:

$$\langle \mathcal{A}, \mathcal{H}\mathcal{A} \rangle = \int_{\bar{\psi}\psi,\bar{\phi}\phi} \mathcal{A}^{\dagger}_{\psi}\mathcal{H}_{\bar{\psi}\phi}\mathcal{A}_{\bar{\phi}} = \sum_{IJ} A^*_I H_{IJ} A_J \sigma_J$$

$$= \sum_{IJ} A^*_I (H^*_{JI}\sigma_I\sigma_J) A_J \sigma_J = \left( \sum_{IJ} A^*_J H_{JI} A_I \sigma_I \right)^* = \langle \mathcal{A}, \mathcal{H}\mathcal{A} \rangle^*, \tag{A.36}$$

for all $\mathcal{A}_{\bar{\psi}} \in \bar{\mathfrak{v}}$.

A Grassmann matrix is said to be *unitary* if it is its own inverse:

$$\int_{\bar{\phi}\phi} \mathcal{U}^{\dagger}_{\bar{\psi}\phi}\mathcal{U}_{\bar{\phi}\psi} = \mathcal{I}_{\bar{\psi}\psi}, \tag{A.37}$$

$$\int_{\bar{\psi}\psi} \mathcal{U}_{\bar{\phi}\psi}\mathcal{U}^{\dagger}_{\bar{\psi}\phi} = \mathcal{I}_{\bar{\phi}\phi}, \tag{A.38}$$

where the *Grassmann identity matrix* is given by

$$\mathcal{I}_{\bar{\psi}\phi} \equiv \sum_I \sigma_I \bar{\psi}^I \phi^I. \tag{A.39}$$

It is easy to check that, despite its unusual form, $\mathcal{I}$ is an identity under the Grassmann matrix multiplication.

## A.6 Parallelism with non-Grassmann linear algebra

So far, all definitions in terms of the coefficients are not very intuitive. However, if we write the coefficient in the right format, the connection with the non-Grassmann linear algebra becomes clear. Let us define the *standard format* of the coefficient tensor to be the one we have been using so far (see (A.25)):

$$\mathcal{T}_{\psi_1 \cdots \psi_m \bar{\phi}_1 \cdots \bar{\phi}_n} = \sum_{I_1, I_2 \cdots, J_n} T_{I_1 \cdots I_m J_1 \cdots J_n} \psi_1^{I_1} \cdots \psi_m^{I_m} \bar{\phi}_1^{J_1} \cdots \bar{\phi}_n^{J_n} \tag{A.40}$$

The *matrix format*, on the other hand, is defined by

$$T^{(\mathrm{m})}_{I_1 \cdots I_m J_1 \cdots J_n} \equiv T_{I_1 \cdots I_m J_1 \cdots J_n} \sigma_{J_1} \cdots \sigma_{J_n}, \tag{A.41}$$

where we multiply the sign factor $\sigma_{J_a}$ for every conjugated index $\bar{\phi}_a^{J_a}$. The coefficient expansion in the matrix format thus becomes

$$\mathcal{T}_{\psi_1 \cdots \psi_m \bar{\phi}_1 \cdots \bar{\phi}_n} = \sum_{I_1, I_2 \cdots, J_n} T^{(\mathrm{m})}_{I_1 \cdots J_n} \sigma_{J_1} \cdots \sigma_{J_n} \psi_1^{I_1} \cdots \psi_m^{I_m} \bar{\phi}_1^{J_1} \cdots \bar{\phi}_n^{J_n}. \tag{A.42}$$

In this format, the Grassmann matrix multiplication can be done in a trivial way. For example, the coefficient matrix $C^{(\mathrm{m})}$ of

$$\mathcal{C}_{\bar{\psi}\phi} = \int_{\bar{\xi}\xi} \mathcal{A}_{\bar{\psi}\xi} \mathcal{B}_{\bar{\xi}\phi} \tag{A.43}$$

can be shown to be equal to the regular matrix multiplication between $A^{(\mathrm{m})}$ and $B^{(\mathrm{m})}$, without any sign factor:

$$\begin{aligned}
\mathcal{C}_{\bar{\psi}\phi} &= \sum_{I,J,K,L} \int_{\bar{\xi}\xi} (A^{(\mathrm{m})}_{IJ} \sigma_I \bar{\psi}^I \xi^J)(B^{(\mathrm{m})}_{KL} \sigma_K \bar{\xi}^K \phi^L) \\
&= \sum_{I,L} \underbrace{\sum_J A^{(\mathrm{m})}_{IJ} B^{(\mathrm{m})}_{JL}}_{= C^{(\mathrm{m})}_{IL}} \sigma_I \bar{\psi}^I \phi^L.
\end{aligned} \tag{A.44}$$

Hermitian conjugation of different objects is now in the intuitive form:

$$\mathcal{A}_\psi = \sum_I A_I^{(\text{m})} \psi^I \longrightarrow \mathcal{A}_{\bar\psi}^\dagger = \sum_I A_I^{(\text{m})*} \sigma_I \bar\psi^I, \tag{A.45}$$

$$\mathcal{B}_{\bar\psi} = \sum_I B_I^{(\text{m})} \sigma_I \bar\psi^I \longrightarrow \mathcal{B}_\psi^\dagger = \sum_I B_I^{(\text{m})*} \psi^I, \tag{A.46}$$

$$\mathcal{M}_{\bar\psi\phi} = \sum_{I,J} M_{IJ}^{(\text{m})} \sigma_I \bar\psi^I \phi^J \longrightarrow \mathcal{M}_{\bar\phi\psi}^\dagger = \sum_{I,J} \underbrace{M_{IJ}^{(\text{m})*}}_{= M_{JI}^{(\text{m})\dagger}} \sigma_J \bar\phi^J \psi^I. \tag{A.47}$$

Hermiticity condition (A.35) in the matrix format now takes the familiar form

$$H_{IJ}^{(\text{m})} = H_{JI}^{(\text{m})*} = H_{IJ}^{(\text{m})\dagger}. \tag{A.48}$$

And the coefficient matrix of the Grassmann identity matrix (A.39) is simply the identity matrix

$$\mathcal{I}_{\bar\psi\phi} = \sum_{I,J} I_{IJ}^{(\text{m})} \sigma_I \bar\psi^I \phi^J \tag{A.49}$$

with $I_{IJ}^{(\text{m})} = \mathbb{1}_{IJ}$.

## A.7    Tensor decomposition

Tensor decomposition is an important operation in tensor network computation. It gives us a way to approximate a large tensor by smaller tensors with lower ranks. In the case of a Grassmann matrix, the Grassmann singular value decomposition (gSVD) is a tensor decomposition of the form

$$\mathcal{M}_{\bar\psi\phi} = \int_{\bar\xi\xi,\bar\zeta\zeta} \mathcal{U}_{\bar\psi\xi} \Sigma_{\bar\xi\zeta} \mathcal{V}_{\bar\zeta\phi} \tag{A.50}$$

where $\mathcal{U}$ and $\mathcal{V}$ are unitary matrices and

$$\Sigma_{\bar\xi\zeta} = \sum_I \lambda_I \sigma_I \bar\xi^I \zeta^I \tag{A.51}$$

is the singular value matrix with $\lambda_I$ being the positively-valued singular value. If $\mathcal{M}$ is Hermitian, the eigenvalue decomposition (gEigD) gives $\mathcal{U} = \mathcal{V}^\dagger$ and $\lambda_I$ being the eigenvalues.

Deriving both the gSVD and gEigD becomes trivial in the matrix format, where we have to perform the non-Grassmann counterpart of the decomposition on the coefficient matrix $M^{(\text{m})}$ to obtain the unitary matrices and the singular value matrix. However, if the Grassmann matrix is Grassmann even; i.e., $\bar\psi^I \phi^J$ is Grassmann even, both of the indices

must have the same parity. This means that the matrix can be diagonalized into even and odd blocks:

$$M = \begin{pmatrix} M^{\mathrm{E}} & \\ & M^{\mathrm{O}} \end{pmatrix}. \tag{A.52}$$

The matrix decomposition can then be performed on the two blocks separately, and we can combine the result into one block in the final step. For the decomposition of the tensor of arbitrary rank, we have to join the legs so that the tensor becomes a matrix first, then we can split the legs after the decomposition.

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
