# Peer review of "GrassmannTN: a Python package for Grassmann tensor network computations"

_SciPost Physics Codebases_

## Round 2 · Referee Report · Bram Vanhecke (Referee 1) · 2023-10-19

Strengths
- Many examples
- Clear explanation from the basics up
Report
A thorough description of a toolbox for Grassmann tensor networks. Plenty of examples and a decent explanation of the underlying theory.

Author: Atis Yosprakob on 2023-10-25 [id 4063]
(in reply to Report 2 on 2023-10-24)I agree with comments (i) and (ii) and will clarify them in the revised version.
For comment (iii), I tried to say that the eigendecomposition of a Hermitian matrix takes the same form as that of the SVD and did not mean to make a direct association between the two. I will rephrase that sentence in the revised version.

---

## Round 2 · Referee Report · Anonymous (Referee 2) · 2023-10-24

Strengths
1- Very clear explanation of Grasmann tensors and their tensor network operations 2- Implementation of necessary basic operations in Grasmann tensor network calculations 3- Sufficient examples to show usefull features of GrassmannTN
Report
This manuscript introduces a Python package for the Grassmann tensor operations. This package supports the complicated fermion sign calculations often appearing in the Grassmann tensor network calculations and provides a sophisticated interface for these functionalities. The author also presents examples of codes for the tensor renormalization group (TRG) and calculations of isometries necessary for many tensor network algorithms.
Grasmann tensors are useful for treating fermionic degrees of freedom in tensor network algorithms. However, when we implement it in program codes, as the author clearly explained, treating signs due to the anti-commuting nature of the fermions becomes complicated. This program package supports the treatment of such complex signs, and users can avoid taking care of them explicitly when they develop their codes. As far as I know, there is no similar open-source code for the Grasmann tensor network. Thus, this package will provide an essential basis for developing and implementing Grasmann tensor network algorithms.
The design principle of the package and examples of codes are clearly written, and I expect that developers will easily try this package. In addition to them, the formulation written in the appendix is helpful for beginners in Grasmann tensors. Thus, I recommend the publication of this manuscript and package in SciPost Physics Codebases.
Minor comments:
(i) In the first sentence of page 21, the author state, "To squeeze the legs, we have to rearrange the indices so that the legs to be squeezed
are separated from the others with grassmanntn.einsum()." I feel that the meaning of the term "separated" is slightly ambiguous. If the author adds more explanations, it may help readers.
(ii)
Although the author does not explicitly write, i and j in Eq. (A. 6) seem restricted to 0 or 1. If this is the case, it is better to write such information explicitly.
(iii)
Below Eq. (A. 51), "If M is Hermitian, the eigenvalue decomposition (gEigD) gives U = V† and λ_I being the eigenvalues.". I think that the eigenvalues of a Hermitian matrix can be negative, although the singular values are non-negative. So, I naively feel that the author implicitly assumes another additional condition for M.
Grasmann tensors are useful for treating fermionic degrees of freedom in tensor network algorithms. However, when we implement it in program codes, as the author clearly explained, treating signs due to the anti-commuting nature of the fermions becomes complicated. This program package supports the treatment of such complex signs, and users can avoid taking care of them explicitly when they develop their codes. As far as I know, there is no similar open-source code for the Grasmann tensor network. Thus, this package will provide an essential basis for developing and implementing Grasmann tensor network algorithms.
The design principle of the package and examples of codes are clearly written, and I expect that developers will easily try this package. In addition to them, the formulation written in the appendix is helpful for beginners in Grasmann tensors. Thus, I recommend the publication of this manuscript and package in SciPost Physics Codebases.
Minor comments:
(i) In the first sentence of page 21, the author state, "To squeeze the legs, we have to rearrange the indices so that the legs to be squeezed
are separated from the others with grassmanntn.einsum()." I feel that the meaning of the term "separated" is slightly ambiguous. If the author adds more explanations, it may help readers.
(ii)
Although the author does not explicitly write, i and j in Eq. (A. 6) seem restricted to 0 or 1. If this is the case, it is better to write such information explicitly.
(iii)
Below Eq. (A. 51), "If M is Hermitian, the eigenvalue decomposition (gEigD) gives U = V† and λ_I being the eigenvalues.". I think that the eigenvalues of a Hermitian matrix can be negative, although the singular values are non-negative. So, I naively feel that the author implicitly assumes another additional condition for M.

---

## Editorial Decision

resubmitted